# Introduction of a Power Law Time-Temperature Equivalent Formulation for the Description of Thermorheologically Simple and Complex Behavior

**DOI:** 10.3390/ma15030726

**Published:** 2022-01-19

**Authors:** Linan Qiao, Sven Nagelschmidt, Uwe Herbrich, Christian Keller

**Affiliations:** Division 4—Safety of Storage Containers, Department 3—Containment Systems for Dangerous Goods, Bundesanstalt für Materialforschung und -Prüfung (BAM), 12200 Berlin, Germany; uwe.herbrich@bam.de (U.H.); christian.keller@bam.de (C.K.)

**Keywords:** time-temperature superposition principle, time-temperature equivalent formulation, bended arrow of time

## Abstract

In this work, a conceptual framework is suggested for analyzing thermorheologically simple and complex behavior by using just one approach. Therefore, the linear relation between master time and real time which is required in terms of the time-temperature superposition principle was enhanced to a nonlinear equivalent relation. Furthermore, we evaluate whether there is any relation among well-known existing time-temperature equivalent formulations which makes it possible to generalize different existing formulations. For this purpose, as an example, the power law formulation was used for the definition of the master time. The method introduced here also contributes a further framework for a unification of established time-temperature equivalent formulations, for example the time-temperature superposition principle and time-temperature parameter models. Results show, with additional normalization conditions, most of the developed time-temperature parameter models can be treated as special cases of the new formulation. In the aspect of the arrow of time, the new defined master time is a bended arrow of time, which can help to understand the corresponding physical meaning of the suggested method.

## 1. Introduction

In many physical and engineering applications the fundamental need exists in terms of extending the experimental region—e.g., prediction of long-time properties from tests conducted in a shorter time range. Besides the time span, further independent influencing variables can be relevant as well, for example, stress/strain, ambient pressure, or radiation. Instead of time dependence, a frequency dependence can occur such as the frequency dependence of viscoelastic moduli of polymers. To extend the experimental region, usually the process temperature is changed and so processes proceed accelerated or decelerated. Further accelerating variables are applied to maintain short time ranges in experiments, for example, use rate, voltage, radiation, or pressure, see Escobar and Meeker [1].

In the field of thermorheology, time and temperature are the two fundamental variables and a state variable—e.g., strain, rupture- and yield-stress of materials, is usually described by a function f(t,ϑ), where t represents the time and ϑ represents the absolute temperature. Regarding the effect of temperature, for nonmetallic materials with viscoelastic behavior (for example for polymers and elastomers), the time-temperature dependence is often described using the time-temperature superposition principle (TTS), which has been developed since the 1950s, see Ferry [2], Schwarzl and Staverman [3] and Findley and Lai [4]. Although the time-temperature superposition principle was developed historically for nonmetallic materials, see Markovitz [5] and Tobolsky and Andrews [6], effective applications of TTS for various classes of materials and processes have been used and confirmed. In a previous work by the authors, the TTS was applied successfully for the description of the relaxation behavior of a type of metal seals, see Qiao, Herbrich and Nagelschmidt [7].

In contrast to the developments of TTS, for metallic materials especially for analyses of fatigue and durability, for example regarding creep-rupture, many time-temperature parameter models have also evolved since the 1950s as representatives for the equivalent effect of time and temperature. In general, a time-temperature parameter model (TTP) considers the two independent variables time and temperature for isostate conditions (f=const.). Therefore, experimental data are depicted and analyzed in the plot of logarithmic time scale versus inverse absolute temperature or absolute temperature to identify equivalent conditions. Indisputably, fundamental works are these of Larson and Miller [8], Manson and Haferd [9], Orr, Sherby and Dorn [10] et al.

To get an overview about the concept, the development and the application of TTP, more detailed studies are recommended, for example, Manson [11], Manson and Halford [12] and Kaufman [13]. Furthermore, major efforts have been made in past decades for a generalization of developed TTP to a ‘single metamodel’, see for example the works of Mendelson, Roberts and Manson [14] and Haque, Ramirez and Stewart [15].

For thermorheologically simple behavior, under isothermal conditions, a change between different but constant temperatures is equivalent to a shift in the logarithmic time scale. Otherwise, for thermorheologically complex behavior, under isothermal conditions, a change between different but constant temperatures is equivalent to a stretch in the logarithmic time scale, see for example the works of Fesko and Tschoegl [16] and Bagley [17]. In this work it is shown that, for the complex behavior, a change between different but constant temperatures is equivalent to a shift and a stretch in the logarithmic time scale.

## 2. Time-Temperature Superposition Principle

The time-temperature superposition principle is generally defined and applied by shifting experimental data along the logarithmic time or frequency axis, whereat one temperature is set as the basis or reference, and, due to the shift, the data range for the reference temperature $\vartheta_{ref}$ will be extended in time or frequency range. As a result of (usually) empirical shifts of data obtained at different but constant temperatures, a composite curve is constructed and often called ‘master curve’ or ‘master function’. An important criterion for the application of TTS is that the shapes of the original curves at different temperatures must match over a substantial range of time or frequencies, see Ferry [2]. Provided that just data and no curves exist, appropriate overlaps of data regarding the investigated state variable for different temperatures must be given.

Based on the assumption that experimental data show thermorheologically simple behavior, an appropriate function of the state variable depending on time and temperature has to be chosen as f(τ,ϑ), for example, representing creep or relaxation, where τ is the normalized real time to a reference time $t/t_{ref}$. This assumption must be confirmed by evaluating the individual results. Generally, thermorheologically simple behaviour is characterized by the fact that functions of the state variable for two arbitrary chosen temperatures can be superimposed by a pure translation—i.e., parallel shifting along the logarithmic time scale. The formulation of a master function is realized variously, either by defining a functional approach, recommended by the authors and considered in this work, or by a user-dependent manual shifting procedure of recorded data obtaining a nearly continuous course, which is often exercised. For user-dependent manual shifting procedures or otherwise nondefined shifts usually a polynomial function is chosen and fitted to the shifted data.

Mathematically, thermorheologically simple behavior is defined for isothermal and isostate conditions by
(1)$$f\left(\tau ,\vartheta \right)=f_{\vartheta_{ref}}\left(\hat{\tau }\right)$$
with the relation between time, temperature and master time $\hat{\tau }$ in the form
(2)$$\hat{\tau }=\alpha \left(\vartheta \right)\tau ,$$
or in logarithmic form
(3)$$ln\left(\hat{\tau }\right)=ln\left(\alpha \left(\vartheta \right)\right)+ln\left(\tau \right).$$

The unknown temperature function α(ϑ) is considered as scale factor in time due to Equation (2) and shift factor in logarithmic time due to Equation (3). As can be seen, the TTS describing thermorheologically simple behavior requires an initial temperature-independent state variable $\partial_{\vartheta }{f|}_{\tau =0}=0$.

For an arbitrarily chosen reference temperature $\vartheta_{ref}$ within the range of minimum and maximum temperature $\vartheta_{min}$ and $\vartheta_{max}$, the corresponding function $f_{\vartheta_{ref}}$ is the so-called master function. That means, the studied process is accelerated or decelerated between $\vartheta_{min}$ and $\vartheta_{max}$. This function is embedded in the function f(τ,ϑ) for $\vartheta \equiv \vartheta_{ref}$
(4)$$f_{\vartheta_{ref}}\left(\hat{\tau }\right)=f\left(\tau ,\vartheta_{ref}\right)$$
with the normalization condition
(5)$$\alpha \left(\vartheta_{ref}\right)=1$$
or
(6)$$ln\left(\alpha \left(\vartheta_{ref}\right)\right)=0,$$
respectively. In addition to the knowledge of a suitable master function $f_{\vartheta_{ref}}\left(\hat{\tau }\right)$, a suitable relation is needed to describe the time-temperature shift factor α(ϑ) for the application of the time-temperature superposition principle. In this regard, the authors suggested a modified Arrhenius approach in an earlier publication, see Qiao, Nagelschmidt and Herbrich [18]. In contrast to the original linear relation of Arrhenius [19] for the temperature-dependent shift factor in logarithmic scale ln(α) and the inverse absolute temperature 1/ϑ, a nonlinear relation was derived for a continuous description within the range of minimum and maximum temperature. Another approach to derive a suitable relation to describe the time-temperature shift factor can be based on the transition state theory wherein different modifications of the Arrhenius equation are suggested, see for example the works of Laidler [20] and Flynn [21].

### New Visualization of TTS

In the following, the applicability of TTS for thermorheologically simple behavior is visualized schematically and explained in detail in order to substantiate the new time-temperature equivalent formulation introduced afterwards. Therefore, an appropriate function was chosen for the description of a state variable f(τ,ϑ). In Figure 1a the projections of the function f(τ,ϑ) are shown for two different temperatures $\vartheta_{ref}$ and $\vartheta_{i}$ over τ, where $\vartheta_{i}$ denotes a temperature between $\vartheta_{min}$ and $\vartheta_{max}$, which was chosen to be above the reference temperature $\vartheta_{ref}$. Furthermore, two isostates $f_{a}$ and $f_{b}$ are selected exemplarily in Figure 1a. Hence, four intersection points can be derived corresponding to $\tau_{1}$ up to $\tau_{4}$. Although a logarithmic time scale is not depicted in Figure 1, it must be pointed out that the distance between the markers (circle and triangle) is identical in logarithmic time scale and therefore, this requirement for thermorheologically simple behavior is fulfilled.

In Figure 1b, the homogeneous linear relations between normalized real time τ and master time $\hat{\tau }$ with the scale factor $\alpha \left(\vartheta_{i}\right)$ according to Equation (2) are depicted for the temperatures $\vartheta_{ref}$ and $\vartheta_{i}$. As can be seen from Figure 1b, one value of master time represents a projection of different values of real time for each corresponding temperature, for example, for ${\hat{\tau }}_{a}$ in the form
(7)$${\hat{\tau }}_{a}⟼\tau_{1}=\frac{1}{\alpha \left(\vartheta_{i}\right)}{\hat{\tau }}_{a}$$
and
(8)$${\hat{\tau }}_{a}⟼\tau_{3}=\frac{1}{\alpha \left(\vartheta_{ref}\right)}{\hat{\tau }}_{a}={\hat{\tau }}_{a}$$

As a fundamental goal of TTS, and depicted in Figure 1c, the function f of the two independent variables τ and ϑ should be represented by only one reduced variable, as exemplarily the master time $\hat{\tau }$ in this work. Due to the reduction of the variables, each isostate condition is defined just by one intersection point on the master-curve—i.e., the triangle marker $\left\{{\hat{\tau }}_{a},\;f_{a}\right\}$ and the circle marker $\left\{{\hat{\tau }}_{b},\;f_{b}\right\}$. On the master function, a given value of $\hat{\tau }$ corresponds to different values for pairs of $\left\{\tau_{k},\;\vartheta_{i}\right\}$. In Figure 1c, the master function $f_{\vartheta }{}_{ref}\left(\hat{\tau }\right)$ is shown based on the reference temperature $\vartheta_{ref}$. For any other temperature $\vartheta_{i}>\vartheta_{ref}$ the function $f\left(\tau ,\vartheta_{i}\right)$ is stretched onto the master function using $\alpha \left(\vartheta_{i}\right)$.

## 3. New Time-Temperature-Equivalent Formulation

In the case of thermorheologically complex behavior, wherein a pure translation along the logarithmic time scale (see TTS) is insufficient in terms of superimposing the function $f\left(\tau ,\vartheta_{i}\right)$ onto the master function, a new formulation is needed. Nevertheless, it is possible to adjust each temperature-dependent factor $\alpha \left(\vartheta_{i}\right)$ so that each superimposed function $f\left(\tau ,\vartheta_{i}\right)$ matches the master function at least in one point, for example at $\left\{{\hat{\tau }}_{a},\;f_{a}\right\}$, see Figure 1c.

To achieve a sufficient match for the projection of a function $f\left(\tau ,\vartheta_{i}\right)$ onto $f_{\vartheta }{}_{ref}\left(\hat{\tau }\right)$ various approaches can be developed. Already in the 1970s, the question of how to deal with thermorheologically complex behavior was discussed, see Fesko and Tschoegl [16]. In this early publication, the authors suggested the construction of a master function for the state variable at the reference temperature with a temperature- and additionally time-dependent shift factor ln(α). However, a specific function of ln(α(ϑ,τ)) was not given. In contrast, in the present work α(ϑ) is considered as a pure temperature-dependent function as for TTS. Additionally, a further temperature-dependent parameter β(ϑ) is introduced here. With this regard, in the following a new formulation is presented for analyzing thermorheologically simple and complex behavior.

### 3.1. Key Assumptions

For a determinate and user-independent procedure for the construction of a master function using the new formulation, some basic or key assumptions are required. Among the following assumptions, fundamentally, the authors recommend to define and to adapt a functional approach based on experimental observations as well as further considerations before shifting any data.

(A-1) For any constant temperature $\vartheta_{i}$, the function of the state variable must be strictly monotonic with respect to time—i.e., for strictly monotonic increasing
(9)$$∆f\cdot ∆\tau >0\;\;\forall \;\left\{∆\tau ,\vartheta_{i}\right\}$$
or for strictly monotonic decreasing
(10)$$∆f\cdot ∆\tau <0\;\;\forall \;\left\{∆\tau ,\vartheta_{i}\right\}.$$

(A-2) The master function, for example, $\tilde{f}\left(\tilde{\tau }\right)$ is defined as a ‘collection’ of all transformed limited functions depending on the reduced variable, whereat, the reduced variable $\tilde{\tau }$ is a function of time and temperature.
(11)$$\left\{f\left(\tau ,\vartheta_{i}\right)\right\}⟼\left\{\tilde{f}\left(g_{i}\left(\tau ,\vartheta \right)\right)\right\}=:\tilde{f}\left(\tilde{\tau }\right)$$

In this work, as a special case, the master function is defined for an arbitrary chosen reference temperature as $f_{\vartheta_{ref}}\left(\tilde{\tau }\right)$ and the reduced variable $\tilde{\tau }$ is referred to as master time.

(A-3) For any constant temperature $\vartheta_{i}$, the master time must be strictly monotonic with respect to the real time.
(12)$$∆\tilde{\tau }\cdot ∆\tau >0\;\;\forall \;\left\{∆\tau ,\vartheta_{i}\right\}$$

### 3.2. New Formulation

Based on the homogenous linear relation of the real time τ and master time $\hat{\tau }$ used for TTS, a new, homogenous nonlinear formulation for the function of the master time $\tilde{\tau }$ was developed in compliance with the key assumptions presented above. In the new formulation the power law is applied for the description of the relation between master time and real time and has the form
(13)$$\tilde{\tau }=\alpha \left(\vartheta_{i}\right)\tau^{\beta \left(\vartheta_{i}\right)},$$
or in logarithmic time scale
(14)$$ln\left(\tilde{\tau }\right)=ln\left(\alpha \left(\vartheta_{i}\right)\right)+\beta \left(\vartheta_{i}\right)ln\left(\tau \right).$$

The normalization conditions to fulfil the key assumption (A-2) are as follows:(15)$$\alpha \left(\vartheta_{ref}\right)=1$$
and
(16)$$\beta \left(\vartheta_{ref}\right)=1,$$
where $\alpha \left(\vartheta_{i}\right)$ denotes the scale factor and $\beta \left(\vartheta_{i}\right)$ represents a form or shape factor in the relation between master time and normalised real time. The last-mentioned factor changes the relation of the master time to real time from linear to nonlinear considering thermorheologically simple versus thermorheologically complex behavior. Obviously, the master time $\hat{\tau }$ used for TTS is a special case of the master time $\tilde{\tau }$ with 𝛽 = 1. The new formulation describing thermorheologically simple and complex behaviors requires also an initial, temperature-independent state variable $\partial_{\vartheta }{f|}_{\tau =0}=0\;$.

As mentioned above, other formulations can be developed as well. Nevertheless, the presented formulation, Equations (13) and (14), can be called the power law time-temperature equivalent principle.

### 3.3. Thermorheologically Complex Behavior

In the following, the applicability of the new introduced TTE formulation for thermorheologically complex behavior is visualized schematically and explained in detail. Therefore, an appropriate function was chosen for the description of a state variable f(τ,ϑ). In Figure 2a, analogous to Figure 1a, the projections of the function f(τ,ϑ) are shown for two different temperatures $\vartheta_{ref}$ and $\vartheta_{i}$ over τ. Furthermore, two isostates $f_{a}$ and $f_{b}$ are selected exemplarily in Figure 2a. Hence, four intersection points can be derived corresponding to $\tau_{1}$ up to $\tau_{4}$. Although a logarithmic time scale is not depicted in Figure 2, it must be pointed out that the distance between the markers (circle and triangle) is not identical in logarithmic time scale in contrast to the requirement for thermorheologically simple behavior.

In Figure 2b, the homogeneous nonlinear relations between normalized real time τ and master time $\tilde{\tau }$ with the scale factor $\alpha \left(\vartheta_{i}\right)$ and the shape factor $\beta \left(\vartheta_{i}\right)$ are depicted according to Equation (13) for the temperatures $\vartheta_{ref}$ and $\vartheta_{i}$. Additionally, one value of master time represents a projection of different values of real time for each corresponding temperature—e.g., for ${\tilde{\tau }}_{a}$ in the form
(17)$${\tilde{\tau }}_{a}⟼\tau_{1}={\left(\frac{1}{\alpha \left(\vartheta_{i}\right)}{\tilde{\tau }}_{a}\right)}^{\frac{1}{\beta \left(\vartheta_{i}\right)}}$$
and
(18)$${\tilde{\tau }}_{a}⟼\tau_{3}={\left(\underset{=1}{\underset{︸}{\frac{1}{\alpha \left(\vartheta_{ref}\right)}}}{\tilde{\tau }}_{a}\right)}^{\underset{=1}{\underset{︸}{\frac{1}{\beta \left(\vartheta_{ref}\right)}}}}={\tilde{\tau }}_{a}.$$

As a fundamental goal of TTE, and depicted in Figure 2c, the state function of the two independent variables τ and ϑ should be represented by only one variable, here the master time $\tilde{\tau }$. Due to the reduction of the variables, one isostate condition is defined just by one intersection point on the master-curve, here the triangle marker $\left\{{\tilde{\tau }}_{a},\;f_{a}\right\}$. On the master function, a given value of $\tilde{\tau }$ corresponds to different values for pairs of $\left\{\tau_{k},\;\vartheta_{i}\right\}$. In Figure 2c, the master function $f_{\vartheta }{}_{ref}\left(\tilde{\tau }\right)$ is shown based on the reference temperature $\vartheta_{ref}$. For a temperature $\vartheta_{i}>\vartheta_{ref}$ the function $f\left(\tau ,\vartheta_{i}\right)$ is stretched onto the master function using $\alpha \left(\vartheta_{i}\right)$ and $\beta \left(\vartheta_{i}\right)$.

## 4. Relation of the New Formulation to Other Time-Temperature Equivalent Parameters

In the field of the TTE formulation several phenomenological parameters have been proposed in the past, for example, by Larson and Miller [8], Manson and Haferd [9] and Orr, Sherby and Dorn [10]. Major efforts have been made for a generalization of different TTP, e.g., by Mendelson, Roberts and Manson [14] and Haque, Ramirez and Stewart [15]. In the aforementioned study of Haque et al., a novel metamodeling approach is applied to derive a single metamodel that incorporates twelve time-temperature parameters (eight existing and four new TTP were exploited). In the following, correlations between the new introduced TTE formulation and exemplarily chosen TTP (Larson–Miller parameter and Haque–Stewart parameter) are shown.

### 4.1. Correlation of the New TTE Formulation to the TTP of Larson and Miller

In the following, the Larson–Miller parameter is presented and compared to the introduced TTE formulation. This parameter is formulated for isothermal and isostate conditions and is given in the form:(19)$$P_{LM}=\vartheta \left[C+ln(\tau \right)]$$

Here, C  is the Larson–Miller constant, and rearrangement of Equation (19) yields a linear relation in the plot of logarithmic time over inverse absolute temperature. If ϑ→∞, there is an intersection point at $\left\{\frac{1}{\vartheta }\to 0,ln\left(\tau \right)=-C\right\}$ for all isostate conditions. Compared with Equation (14) and by setting
(20)$$P_{LM}\equiv ln\left(\tilde{\tau }\right)$$
it formally follows
(21)ϑC=ln(α)
and
(22)ϑ=β.

Applying the normalization condition Equation (15), Equation (21) is satisfied only for $\vartheta_{ref}=0$, and applying the normalization condition Equation (16), Equation (22) is just satisfied for $\vartheta_{ref}=1$, leads to an *inconsistency* for the Larson–Miller parameter in its original form. Therefore, an adjusted Larson–Miller parameter is introduced which satisfies the normalization requirements of the new TTE formulation—i.e., the scale of time must not be changed for the reference temperature. The adjusted form is defined by
(23)$${\tilde{P}}_{LM}=\frac{\vartheta }{\vartheta_{ref}}\left[C+ln(\tau \right)]-C$$
compared with Equation (14) and by setting
(24)$${\tilde{P}}_{LM}\equiv ln\left(\tilde{\tau }\right)$$
it formally follows
(25)$$\left(\frac{\vartheta }{\vartheta_{ref}}-1\right)C=ln\left(\alpha \right)$$
and
(26)$$\frac{\vartheta }{\vartheta_{ref}}=\beta .$$

Applying the normalization condition Equations (15) and (16), Equations (25) and (26) are both satisfied for $\vartheta =\vartheta_{ref}$. Hence, a *consistency* is given for the adjusted, normalized Larson–Miller parameter ${\tilde{P}}_{LM}$ in the form of Equation (23), where C is determined with ln(α)/(β−1). In the Arrhenius plot of logarithmic time over $\vartheta_{ref}/\vartheta$, the slope of the linear relation is given by ${\tilde{P}}_{LM}+C$. If ϑ→∞, the same intersection at $\left\{\frac{1}{\vartheta }\to 0,ln\left(\tau \right)=-C\right\}$ appears for all isostate conditions just as in the original Larson–Miller formulation. That means the principal characteristic of the Larson–Miller parameter is not changed, but the constant C and the parameter ${\tilde{P}}_{LM}$ are dimensionless due to the normalization and especially due to the satisfaction of assumption (A-2).

### 4.2. Correlation of the New TTE Formulation to the TTP of Haque and Stewart

In the following, the more general Haque–Stewart parameter, which formally incorporates twelve different TTP, is presented and compared to the introduced TTE formulation. This parameter is formulated for isothermal and isostate conditions and given in the form
(27)$$P_{HS}=\frac{ln(\tau )-a_{0}-a_{1}\vartheta^{r}}{{\left(\vartheta^{r}-a_{2}^{r}\right)}^{q}}$$
where $a_{0}$, $a_{1}$ and $a_{2}$ are material coefficients, r=±1 and q∈ℕ. Analogous to the performed adjustments for the Larson–Miller parameter, the adjusted Haque–Stewart parameter is introduced in the form
(28)$${\tilde{P}}_{HS}=\frac{a_{0}+a_{1}\vartheta_{ref}^{r}}{{\left(\vartheta_{ref}^{r}-a_{2}^{r}\right)}^{q}}-\frac{a_{0}+a_{1}\vartheta^{r}}{{\left(\vartheta^{r}-a_{2}^{r}\right)}^{q}}+{\left(\frac{\vartheta_{ref}^{r}-a_{2}^{r}}{\vartheta^{r}-a_{2}^{r}}\right)}^{q}ln\left(\tilde{\tau }\right)$$
compared with Equation (14) and by setting
(29)$${\tilde{P}}_{HS}\equiv ln\left(\tilde{\tau }\right)$$
it formally follows
(30)$$\frac{a_{0}+a_{1}\vartheta_{ref}^{r}}{{\left(\vartheta_{ref}^{r}-a_{2}^{r}\right)}^{q}}-\frac{a_{0}+a_{1}\vartheta^{r}}{{\left(\vartheta^{r}-a_{2}^{r}\right)}^{q}}=ln\left(\alpha \right)$$
and
(31)$${\left(\frac{\vartheta_{ref}^{r}-a_{2}^{r}}{\vartheta^{r}-a_{2}^{r}}\right)}^{q}=\beta .$$

By setting $a_{1}=a_{2}=0,\;r=-1,\;q=1$ and $a_{0}=-C/\vartheta_{ref}$ one gets the normalized Larson–Miller parameter ${\tilde{P}}_{LM}$ again. Hence, all the twelve different TTP which are included in the normalized Haque–Stewart metamodel are special cases of the new TTE formulation. Therewith, a more general formulation in the field of TTE is introduced with this study as well.

## 5. The New TTE Formulation Relating to ‘The Arrow of Time’

Considering the new TTE formulation with regard to ‘the arrow of time’, see for instance Wallace [22], leads to results, that the new formulation satisfies appropriate conditions, especially, due to a strictly monotonic characteristic, see assumption (A-3). As mentioned above, other comparable formulations may exist, whereat a verification should be checked for each formulation with the condition that the second law of thermodynamics must be satisfied. In this context, the concept of the arrow of time is useful. Considering an isothermal process at a temperature $\vartheta_{i}$ and the idea of the arrow of time a homologous time or arrow of time $\tau_{\to }$ for normalized real time can be defined in the form
(32)$$\tau_{\to }\left(\tau \right)=\frac{\tau -\tau_{min}}{\tau_{max}-\tau_{min}}\;\in \left[0,1\right]$$
with the period of observation $\tau_{min}<\tau <\tau_{max}$, where $\tau_{\to }$ is a *linear* function of τ. Analogously a homologous time or arrow of master time ${\tilde{\tau }}_{\to }$ for normalized master time can be defined in the form
(33)$${\tilde{\tau }}_{\to }\left(\tilde{\tau }\right)=\frac{\tilde{\tau }-{\tilde{\tau }}_{min}}{{\tilde{\tau }}_{max}-{\tilde{\tau }}_{min}}\;\in \left[0,1\right]$$
with the period of application ${\tilde{\tau }}_{min}<\tilde{\tau }<{\tilde{\tau }}_{max}$.

Using the nonlinear relation between master time and real time, from Equation (13), the arrow of master time ${\tilde{\tau }}_{\to }$ can be rewritten in the following form
(34)$${\tilde{\tau }}_{\to }\left(\tau \right)=\frac{\alpha \tau^{\beta }-{\tilde{\tau }}_{min}}{{\tilde{\tau }}_{max}-{\tilde{\tau }}_{min}}\;\in \left[0,1\right].$$

Whereat, ${\tilde{\tau }}_{\to }$ is a *nonlinear* function of τ, due to the additionally introduced parameter $\beta \left(\vartheta_{i}\right)$. The definitions of the presented ‘homologous time forms’ are schematically depicted in Figure 3. From that, the range of observation, the relation between real time and master time as well as the thus dependent range of application for a temperature $\vartheta_{i}$ can be gathered. Furthermore, in Figure 3 can be seen that the arrow of master time ${\tilde{\tau }}_{\to }$ is bended but goes forward. In contrast, for TTS, the arrow of master time is linear proportional to the arrow of time. That means, the arrow of time remains straight even in the aspect of master time. For both, the second law of thermodynamics is satisfied.

## 6. Conclusions

Regarding the effect of temperature for the description of time-accelerated processes, different strategies have been developed. For thermorheologically simple and complex processes or material behavior the time-temperature superposition principle and enhanced formulations, respectively, are commonly used. In contrast, in the past many time-temperature parameter models as representatives for the equivalent effect of time and temperature have been evolved as well. Both strategies and corresponding formulations are summarized in this work. For the description of both thermorheologically simple and complex behavior by using just one approach, a new power law time-temperature equivalent formulation was introduced in the present work. Further advantages are a continuous description in time as well as a more precise user-independent prediction. The new formulation can be applied for various processes or material behaviors and depending on the field of application, it can be adapted for the definition of an appropriate functional approach for the state function. Hence, the introduced power law time-temperature equivalent formulation appears to be the most general form in the field of time-temperature equivalence up to now. The new formulation was exemplarily related to adjusted forms of well-known time-temperature equivalent parameter models, such as those proposed by Larson and Miller and by Haque and Stewart. In addition, the relation between the real time and master time was discussed, bearing the ‘the arrow of time’ in mind. Considering that, the arrow of master time is bent but goes forward and satisfies the second law of thermodynamics.

## Figures and Tables

**Figure 1 materials-15-00726-f001:**
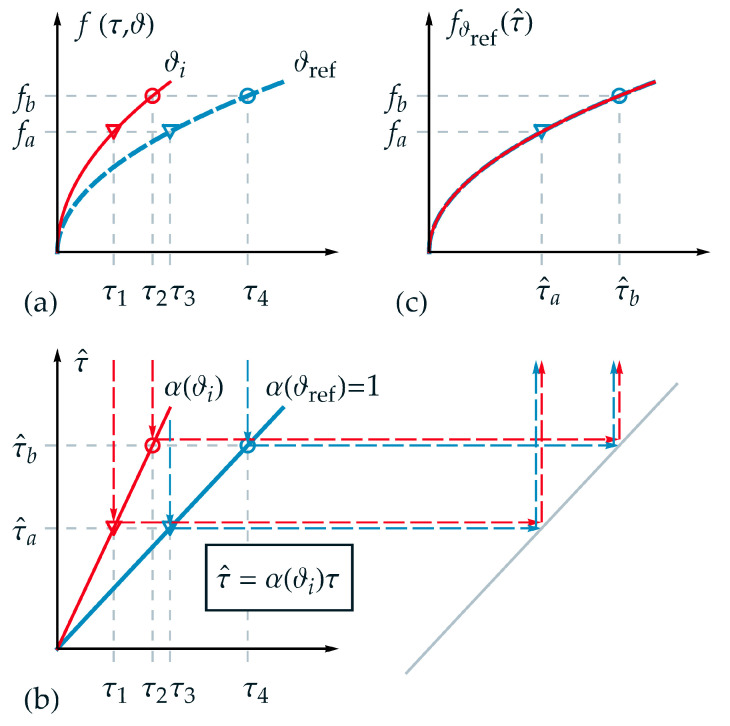
Application of the TTS for thermorheologically simple behavior: (**a**) projections of function f(τ,ϑ) over normalized real time τ for two different temperatures $\vartheta_{ref}$ and $\vartheta_{i}$; (**b**) linear relation between normalized real time τ and master time $\hat{\tau }$; (**c**) master function $f_{\vartheta }{}_{ref}\left(\hat{\tau }\right)$ over master time $\hat{\tau }$ based on the reference temperature $\vartheta_{ref}$ with the transformation of exemplarily chosen normalized real time values: $\tau_{1}⟼{\hat{\tau }}_{a}=\alpha \left(\vartheta_{i}\right)\tau_{1}=\tau_{3}$ and $\tau_{2}⟼{\hat{\tau }}_{b}=\alpha \left(\vartheta_{i}\right)\tau_{2}=\tau_{4}$.

**Figure 2 materials-15-00726-f002:**
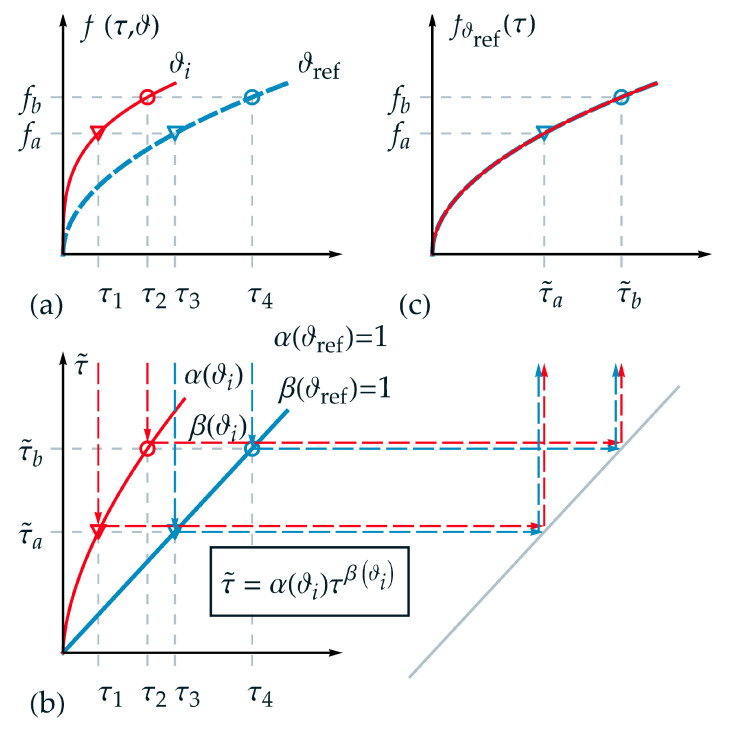
Application of the new formulation for thermorheologically complex behavior: (**a**) projections of function f(τ,ϑ) over normalized real time τ for two different temperatures $\vartheta_{ref}$ and $\vartheta_{i}$; (**b**) nonlinear relation between normalized real time τ and master time $\tilde{\tau }$ for temperature $\vartheta_{i}$; (**c**) master function $f_{\vartheta }{}_{ref}\left(\tilde{\tau }\right)$ over master time $\tilde{\tau }$ based on the reference temperature $\vartheta_{ref}$. For a temperature $\vartheta_{i}>\vartheta_{ref}$ the function $f\left(\tau ,\vartheta_{i}\right)$ is stretched onto the master function using $\alpha \left(\vartheta_{i}\right)$ and $\beta \left(\vartheta_{i}\right)$: $\tau_{1\;}⟼{\tilde{\tau }}_{a}=\alpha \left(\vartheta_{i}\right)\tau_{1}{}^{\beta \left(\vartheta_{i}\right)}=\tau_{3}$ and $\tau_{2\;}⟼{\tilde{\tau }}_{b}=\alpha \left(\vartheta_{i}\right)\tau_{2}{}^{\beta \left(\vartheta_{i}\right)}=\tau_{4}$.

**Figure 3 materials-15-00726-f003:**
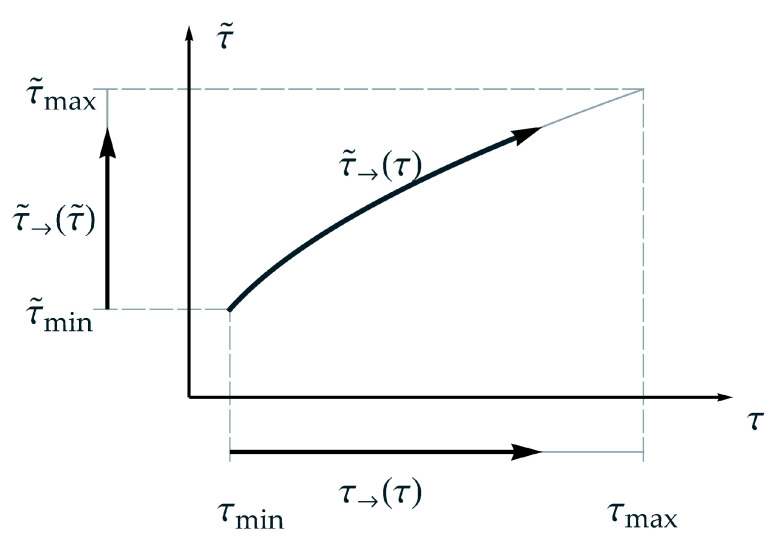
Schematic representation of the suggested nonlinear relation between master time and real time regarding and applying ‘the arrow of time’, respectively.

## Data Availability

No special data were used.

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
