# Peer review of "Introduction of a Power Law Time-Temperature Equivalent Formulation for the Description of Thermorheologically Simple and Complex Behavior"

_materials, 2022, doi:10.3390/ma15030726_

Round 1

Reviewer 1 Report

While the paper appears to intend a new geneeralization of time-temperature superposition (TTS), it remains unclear for which class or classes of material this should be valid. Coming from the polymer field, it is well known that the Arrhenius and Williams-Landel-Ferry (WLF, altrernately VFTH) equations for calculating shift factors in TTS exist side by side, being applicable for different materials in dependence of distance to Tm and Tg (an excellent study presenting an harmonization approach has just been published recently, see https://doi.org/10.1002/prep.201800329). Also, thermorheological complexity practically always has a clear reason, be it a mobility transition, branching structure or phase separation (see e.g. https://doi.org/10.1007/s00397-015-0864-9). It is rather unlikely that the newly proposed "arrow of time" concept (the explanation of which is rather slim, to say the least) can compensate all these possible cases. 

If, on the other hand, the new model should be limited in applicability to metals only, then this should be stated clearly (including a statement for the case of alloys, where multiple phase transitions need to be considered as well). In absence of experimental data, the actual scope of the paper remains otherwise unclear.

Reviewer 2 Report

Dear authors,

I am not educated in physics or material sciences, but in applied geosciences. So, my experience with time – temperature dependence and rheological behaviour is a bit different then it was described in manuscript. However, I could represent one of average reader for this paper. Consequently, here are my comments:

Title is partially misleading with “part I: theory”, except of you do not have arrangement with journal editor to publish part II: practice”. The title must reflect journal scope, not intention used for book chapters.

In introduction is not clear which materials are considered as base for your theoretical development and review. Rheology is very wide discipline regarding medium considered. It could be examined as deformation of particular material. Responding of material property on force can be considered for solid, liquid, or gaseous and depend on shape of flow space. If you put the stress on thermal properties (vs. strain, i.e., shear), where expansion or shrinking of volume took place, it would be welcome to give examples of material that fit your assumed model, especially if you transfer time shift in logarithm scale.

So, consequently it arises the question – did you set up model of viscous or elastic deformation, or mixed viscoelastic? Did you consider fluid as simple as Newtonian (apparent viscosity is constant, i.e., independent of shear rate) or non-Newtonian? Or observed elastic solids? Or I just missed some statements in text?

Figures – are too small and very low quality. Each of them must be associated with (colour) legend and symbol description.

Lines 358-364: “The new formulation can be applied for various processes or material behaviours and depending on the field of application…”. The reader must know on which processes or materials you mean, if you pretend to put new theoretical tool, even paradigm.

Reference list –

I am not sure what is scope of your work (material science or rheology in real (sub)surface conditions), but regarding geosciences there is pretty wide application of time-temperature index, or Lopatin index, (based on power of 2), as tool for rock heating calculation in the past. E.g.,

https://pubs.geoscienceworld.org/aapgbull/article-abstract/64/6/916/37275/Time-and-Temperature-in-Petroleum-Formation?redirectedFrom=PDF

https://www.osti.gov/biblio/5848993-modeling-oil-generation-time-temperature-index-graphs-based-arrhenius-equation

Kind regards,

reviewer

Round 2

Reviewer 1 Report

The corrections have made this paper fully acceptable for publication.

Author Response

Dear Reviewer,

thank you for reviewing our paper. We have revised the draft again with respect to minor spelling changes and punctuation. Furthermore, the draft was adapted to the standard paper template. The figures have been produced again in a higher quality and corresponding to the explanations in the text.

With kind regards

Linan, Uwe, Christian, Sven
